# Contribution of *pks*[+] *E. coli* mutations to colorectal carcinogenesis

Bingjie Chen[1,2,7], Daniele Ramazzotti [ID][3,7], Timon Heide[1,4], Inmaculada Spiteri[1], Javier Fernandez-Mateos [ID][1], Chela James[1,4], Luca Magnani [ID][5,6], Trevor A. Graham [ID][1] [✉] & Andrea Sottoriva [ID][1,4] [✉]

The dominant mutational signature in colorectal cancer genomes is C > T deamination (COSMIC Signature 1) and, in a small subgroup, mismatch repair signature (COSMIC signatures 6 and 44). Mutations in common colorectal cancer driver genes are often not consistent with those signatures. Here we perform whole-genome sequencing of normal colon crypts from cancer patients, matched to a previous multi-omic tumour dataset. We analyse normal crypts that were distant vs adjacent to the cancer. In contrast to healthy individuals, normal crypts of colon cancer patients have a high incidence of *pks + (polyketide synthases) E.coli* (*Escherichia coli*) mutational and indel signatures, and this is confirmed by metagenomics. These signatures are compatible with many clonal driver mutations detected in the corresponding cancer samples, including in chromatin modifier genes, supporting their role in early tumourigenesis. These results provide evidence that *pks + E.coli* is a potential driver of carcinogenesis in the human gut.

Colon carcinogenesis is the archetypal model of step-wise accumulation of malignant traits[1] that, through a process of Darwinian selection for genetic[2] but likely also epigenetic[3] alterations, gives rise to a malignancy. Typical mutations in APC, KRAS, TP53, SMAD4 and other cancer driver genes are found in the large proportion of colorectal tumours[4] and are almost invariably clonal within a cancer[3,5]. Most colorectal cancers are microsatellite stable (MSS) while also being characterised by chromosomal instability (CIN)[6]. A minor subset of cases (15%) carries microsatellite instability (MSI) and genomes relatively devoid of copy number alterations. MSI colon cancers have a much higher mutational load[7] due to deficiency of mismatch repair (MMRd) but share many common driver genes with MSS, and indeed show a similar number of mutations in drivers, despite their higher mutational burden[8].

The most common mutational process in the human colon is the CpG deamination signature (signature 1), causing C > T mutations at methylated CG sites[9]. This signature is also the most common one across the normal colon of many mammal spieces[10]. In MSS cancers, signature 1 remains the dominant mutational footprint, whereas in MSI cases the MMR signatures 6 and 44 become dominant after inactivation of mismatch repair genes[11]. However, these signatures alone do not always explain the specific substitutions in trinucleotide context we observe in genes driving colorectal carcinogenesis, which are often not compatible with these common signatures.

Some strands of *Escherichia coli* can contain the *polyketide synthetase (pks) island* that encodes *colibactin*, a genotoxic compound that can alkylate DNA on adenine residues and induce point mutations with a specific signature[12,13] (COSMIC SBS88). Furthermore, exposure to *pks*[+] *E. coli* generates a characteristic short indel signature (COSMIC ID18) which manifests as short T deletions at T homopolymers[10]. *Pks*[+] *E.coli* has been found in colon cancer[14,15] and the corresponding signature has been detected in cell's genomes in both normal[9] and cancer[12,13]. However, to our knowledge it has not been yet identified in normal colon of cancer patients (suggesting prolonged exposure), as

[1]Centre for Evolution and Cancer, The Institute of Cancer Research, London, UK. [2]GMU-GIBH Joint School of Life Sciences, The Guangdong-Hong Kong-Macau Joint Laboratory for Cell Fate Regulation and Diseases, Guangzhou Medical University, Guangzhou, China. [3]Department of Medicine and Surgery, University of Milano-Bicocca, Milan, Italy. [4]Computational Biology Research Centre, Human Technopole, Milan, Italy. [5]Division of Breast Cancer Research, The Institute of Cancer Research, London, UK. [6]Department of Surgery and Cancer, Imperial College London, London, UK. [7]These authors contributed equally: Bingjie Chen, Daniele Ramazzotti. [✉]e-mail: trevor.graham@icr.ac.uk; andrea.sottoriva@fht.org

comprehensive analyses of matched normal and cancer tissues from the same patient are lacking. Moreover, sampling strategies so far have been limited to single bulk tissue whereas multiple spatial sampling at single clone resolution is important to determine whether *pks*⁺ *E. coli* is only superficial to the colon, forming a film, or pervades the inner epithelium. Finally, analyses on the causative link between *pks*⁺ signatures and driver mutations are missing. For these reasons, the contribution of this process to carcinogenesis and colorectal cancer incidence is largely unknown.

Here we study the mutational signatures in the human gut using single crypt whole-genome sequencing collected from patients with cancer. We compared the genomes of distant normal crypts, normal crypts that are adjacent to the tumour, and cancer glands from the same patients. We find that *pks*⁺ *E.coli* is pervasive in the normal colon of cancer patients and is the candidate process responsible for many mutations in cancer driver genes in colorectal malignancies.

## Results

### Whole-genome profiling of normal colon crypts both distant and adjacent to cancer

We recently profiled a set of primary colorectal cancers at single-gland resolution using a multi-omic strategy combining DNA, chromatin and RNA measurements[3,16]. We refer to this dataset as the "EPICC cohort" (Evolutionary Predictions in Colorectal Cancer). From the same set of patients, in this study we collected normal colon crypts and profiled those with whole-genome sequencing (WGS, Fig. 1A,B). We also collected normal crypts that were adjacent to cancer glands (i.e., mixed up with cancer) to study the field effect of the malignant microenvironment. In this study, we used a total of 366 deep WGS samples from single glands, including 101 normal samples (74 adjacent and 27 distant normal crypts) that were not published before. Another 265 cancer samples used in this study are from matched cancer tissues from the same patient and are part of the original EPICC publications[3,16]. We report the sample naming convention in Fig. 1A (bottom left), where each sample name contains information about the patient (e.g. C519) the region where it was sampled (A, B, C, D are cancer regions, E is the distant normal region) and the sample number within the region (e.g. G1, G2, …, for glands/crypts; B1, B2, …, for bulk samples).

We found that upon qualitative assessment of microscopy images, normal crypts adjacent to the cancer were significantly larger than normal that were distant (Fig. 1C), possibly due to aberrant WNT signalling in the environment of cancer glands[17]. Both distant and adjacent normal contained some driver mutations, as previously reported[9] (Fig. 1D for samples with a mutation, see Supplementary Fig. 1 for all samples), although positive selection for those drivers was detected by dN/dS[8] only for truncating mutations in adjacent normal crypts (Fig. 1E). Distant normal crypts did show some evidence of negative selection (i.e., depletion of mutations) for missense variants in tumour suppressor genes. The mutational burden was also different depending on the location of the normal samples, with adjacent normal having significantly higher mutational burden than distant normal crypts, but still lower than cancer glands (Fig. 1F). At the phylogenetic level, normal crypts including both distant and adjacent ones, were completely distinct from the corresponding cancer, representing independent lineages as previously reported[9,18,19] (Supplementary Fig. 2).

### *Pks*⁺ *E. coli* signatures are pervasive in the normal gut of colon cancer patients

We combined the data from normal and neoplastic samples from our cancer patients with a recent dataset from normal crypts of healthy patients[9], and performed mutational signature discovery using SparseSignature[20], a method that avoids overfitting of signatures by enforcing sparsity of the signal (Supplementary Fig. 3). We obtained signatures with very high similarity to COSMIC signatures (see

Supplementary Fig. 3 for reported cosine similarities) while ensuring we were not overcalling. We detected the presence of the *pks*⁺ *E.coli* signature (signature SPS7 in our analysis, corresponding to COSMIC SBS88 – see Supplementary Figs. 3 and 4). The *pks*⁺ signature was observed only in a small proportion of crypts from healthy individuals, but in the majority of normal crypts from cancer patients in our cohort (Fig. 2A and Supplementary Fig. 4). The signature was also present in the corresponding cancer samples in similar proportion, suggesting that crypts with high *pks*⁺ signature may be vulnerable to tumorigenesis. The same signature, when present, was lower in normal crypts from healthy patients (Fig. 2B). Specifically, in 23/30 patients we found evidence of *pks*⁺ signature in the cancer (with normalised exposure >5%). Out of the 17 patients for which we had adjacent normal crypts, *pks*⁺ signature was detected in those normal tissues in 12 cases. Out of the 10 patients with distant normals, half of them had *pks*⁺ signature in those samples. We found the *pks*⁺ signature in 76.67% of cancer crypts, 70.59% for adjacent normal, and 50% for distant normal from cancer patients compared to 30% (12/40) in individuals without cancer (Supplementary Data 1). This result revealed that the *pks*⁺ signature is strongly enriched in the normal colon of colorectal cancer patients compared to healthy.

We found similar proportion of the *pks*⁺ signature in clonal vs subclonal mutations while higher proportion of it in normal crypts from cancer patients (Fig. 2B). Due to the abundancy of other signatures in cancer samples, the *pks*⁺ signature appeared more diluted in cancer, whereas it is a dominant source of mutations in the normal colon. Indeed, in our cancer cohort we did not find the *pks*⁺ *E.coli* signature in our original analysis[3]. To elucidate this, we performed de novo signature identification separately in distant normal samples, adjacent normal samples, and repeated the cancer-only analysis of our original publication. We indeed were able to identify the *pks*⁺ *E.coli* signature in both normal cohorts (Supplementary Fig. 5A, B), but not in the cancer samples alone, where the dominant signatures were those reported previously[3] (Supplementary Fig. 5C). We then analysed the cancer cohort more in depth, and compared the cosine similarities of signature extraction with and without the *pks*⁺ *E.coli* signature, hence assigning only SPS1-6 versus assigning SPS1-7. The exclusion of SPS7 resulted in a significant decrease in the goodness of fit (Wilcoxon test *p* value = 0.033, Supplementary Fig. 5D). These results provide evidence for the significant impact of SPS7 on the mutations observed in cancer patients as well, highlighting the difficulties of inferring robust mutational signatures. In our case, the addition of the normal samples from the same patient of the EPICC cancer cohort, as well as the normal samples from the Lee-Six et al.[9] (2019), allowed for the identification of the *pks*⁺ signature in our cancer cohort too.

On top of the single base signature discussed so far, it is known that colibactin from *pks*⁺ *E.coli* also induces short deletions at T homopolymers[9,13]. We investigated whether short T deletions at T homopolymers occurred in both normal and cancer crypts of our dataset. The short T-del signature was even more evident than the single base *pks*⁺ signature (SPS7/SBS88) and could be found in nearly all samples (Fig. 2C). When comparing the EPICC normal and cancer samples, both distant and adjacent normal showed even higher contribution of the short T-del signature in normals than cancer, again likely to additional indel signatures in tumours (Fig. 2D). Moreover, the amount of SPS7 signature and the short T-del signature per sample were correlated both for clonal (Fig. 2E) and subclonal mutations (Fig. 2F).

### *Pks*+ *E.coli* genomic DNA can be detected in the same samples

We then performed metagenomic analysis on the same samples to find the presence of reads from *pks*⁺ genes. As shown in Fig. 3, the *pks*⁺ genes could be detected in the sequencing data from the samples (either cancers or normal) in 19 patients out of 30, thus validating the presence of the mutational and indel signatures (see also Supplementary Figs. 6 and 7). Notably, whereas the presence of *pks*⁺ genes in

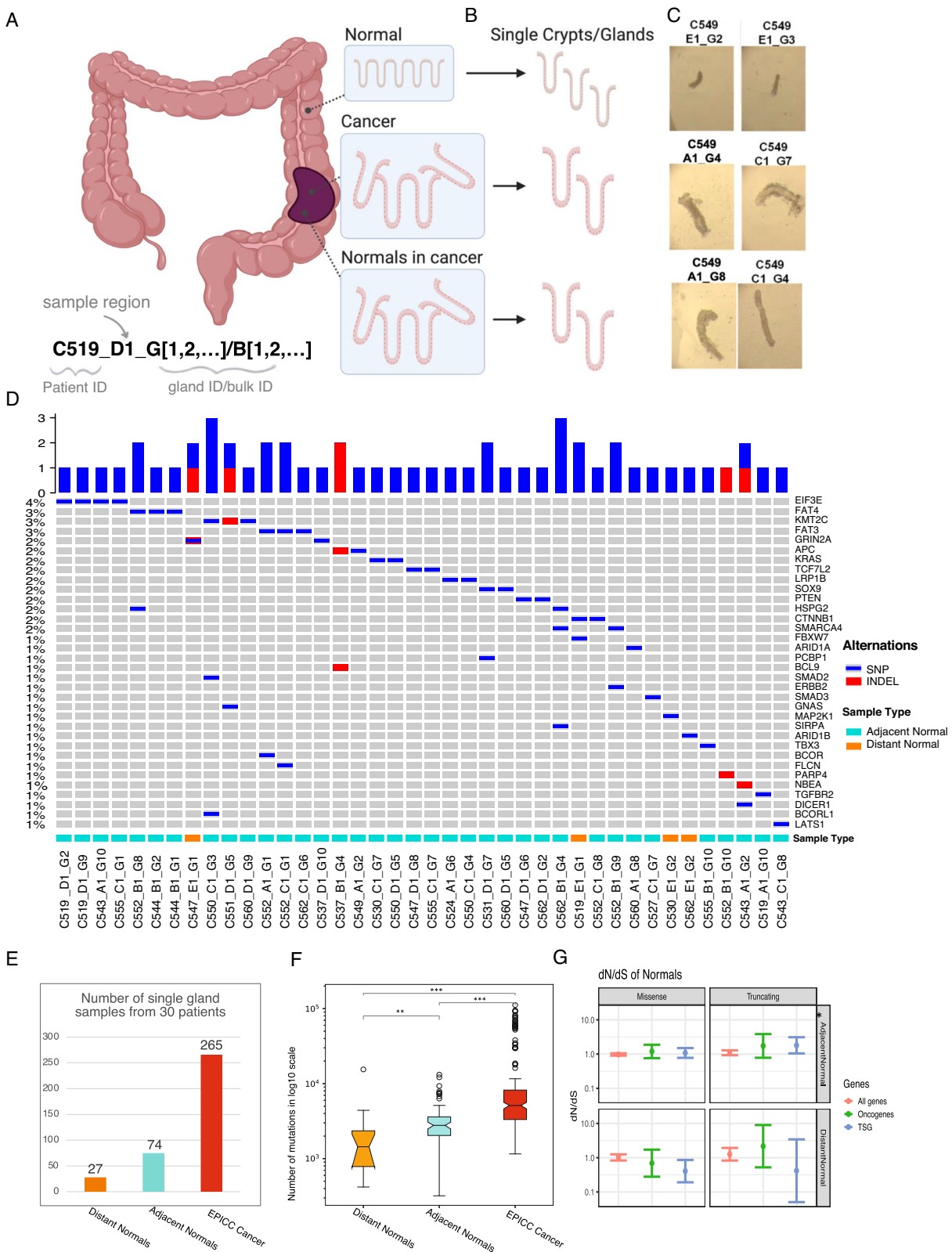

the sample indicates the presence of the genotoxic *E. coli* strands, the mutational signature is the result of mostly past exposure, hence it could be accumulated in cell genomes in the past without *pks*+ being present at the moment of sampling. Despite this, we found the proportion of samples with *pks*+ in MSS patients to be positively correlated with the proportion of short T-dels (Supplementary Fig. 8, R2 ~ 0.77, *p* = 3.6e-7). In MSI cases instead, which are dominated by a

longer T-del signature, the signal was diluted. Significantly more short T-dels were observed in samples with presence of *pks*+ *E.coli* metagenomic reads (Supplementary Fig. 9A). The same difference was not evident for the base substitution signature, likely due to dilution by other signatures (Supplementary Fig. 9B), although a trend was noticeable when focussing on T > C in ATT context (Supplementary Fig. 9C).

**Fig. 1 | Study design. A** Cancer and paired normal samples were collected from fresh colectomy specimens from 30 colorectal cancer patients. Normal colon samples (denoted as distant normal in below) are from the distant epithelium which are several centimetres away from tumour lesions, the normals in cancer (marked as adjacent normal) are the normal crypts isolated from the tumour tissues. Sample naming convention is also reported. **B** Single glands were dissected from normal and cancer samples, and we performed Whole-genome sequencing for these samples. **C** We collected individual crypts from normal and cancer samples and performed qualitative morphology examination, the crypts (both adjacent normal and cancer) from different regions of cancer tissue were marked as 'A_','B_','C_','D_', while 'E_' are the crypts from distant normal tissues. **D** Oncoprint of mutations in colorectal cancer driver genes in normal crypts in our cohort (only samples containing at least one mutation in a driver gene are included). **E** Number of single gland samples from the 30 patients. **F** Mutational burden for each set of samples, the sample size of each group are showed in (**E**). In the boxplots of panels, hinges indicate the 25th, 50th, and 75th percentiles, whiskers indicate 1.5 × interquartile ranges, and dots indicate values of individual samples. Two-sided Mann–Whitney analysis was applied to compare groups. **G** dN/dS analysis for measuring selection on driver mutations.

### *Pks*+ *E.coli* mutational and indel signature explains mutations in driver genes of matched cancers

Finally, we investigated the contribution of *pks*+ signatures to the mutation of common driver genes and chromatin modifier genes (cmgs) in cancers from our EPICC cohort. The exposure matrix and the signatures matrix obtained with the de novo signatures inference of *SparseSignatures* allows to compute an expected probability for a given trinucleotide context to be mutated per patient. We note that signatures are categorical distributions over the 96 trinucleotides context, and the probability of a signature causing a specific mutation for a patient is the probability of mutating such trinucleotide context given the signature, normalised for the number of mutations the signature is generating for the patient (alpha matrix). Hence, we can assign an expected probability for a given mutation to be generated for each signature. This model assumes a uniform signatures activity over time.

This analysis revealed that, besides SPS1 (COCMIS signature SBS1, clock-like deamination) and SPS3 (COSMIC signature SBS5 clock-like signature), the *pks*+ mutational signature is another important factor which may have caused mutations in driver genes and chromatin modifier genes (Fig. 4). Notably, in patients C530, C531, C544, C547 and C550, it is highly likely that the *pks*+ signature caused key driver mutations, including stereotypical drivers of CRC such as APC, KRAS and PIK3CA (Fig. 4A). We have previously shown that mutations in chromatin modifier genes in colorectal cancer are under positive selection[16], suggesting their role in tumourigenesis as possible contributors to epigenetic instability. We found that a large proportion of mutations in chromatin modifier genes in MSS cancers are caused by the *pks*+ signature (Fig. 4B). The same analysis is reported in Supplementary Fig. 10 for MSI cancers, where the dominant causative signature of driver and chromatin gene mutations is instead SPS6 (mismatch repair signature COCMIS SBS44) as expected. Furthermore, we report that multiple alterations in cancer driver gene and chromatin modifier genes are consistent with *pks*+ induced short T-dels (Fig. 4C). These results paint a picture of *pks*+ as a causative factor of DNA alterations in genes involved in cancer. Those alterations would be hard to explain with other common mutational processes acting on the genome, such as COSMIC signature 1 or MMR signatures.

The EPICC study also included pre-cancerous adenomas (polyps). We evaluated the *pks*+ signatures for the most common driver mutations (APC) in the polyps of patient C561. And, as shown above, we found both *pks*+ signature and *pks*+ *E.coli* reads in C561. In Fig. 5, the APC mutations in samples from polyps also match the *pks*+ signature. In polyp G, we detected a T to C mutations in ATT context on APC gene and in polyp F, there is a short T-del in T-homopolymer. It indicates that *pks*+ may be a candidate process responsible for mutations in cancer driver genes in this cancer patient.

### Discussion

Colorectal cancer is one of the most common adult malignancies. Moreover, this type of tumour has one of the fastest increasing incidences in adults under 40 years old, and nobody knows why[21]. Colorectal carcinogenesis is clearly linked to ageing of the cells in the gut as the incidence in the general population drastically increases with age.

Moreover, the mutational signature that is most prevalent in colon cancers is COSMIC Signature 1, representative of C > T deamination at CpG sites in the ageing genome. However, such signature is not consistent with many driver mutations we find in colon cancers.

Here, leveraging on a unique matched sample set of distant normal, adjacent normal and cancer, we investigate the prevalence of *pks*+ genotoxic *E.coli* as a possible contributor to colorectal cancer tumourigenesis. This strain of *E.coli* has been demonstrated to be genotoxic experimentally and has also been found in large cohorts[22].

Seminal studies have identified the presence of *pks*+ *E.coli* in normal and cancer intestinal tissues[13,15], and characterized its functional consequences on the cell's genome. In this study we leveraged a unique clinical dataset that combines regionally separated normal colonic tissues from cancer patients and their matched malignancy using whole-genome sequencing. We contrasted the prevalence of *pks*+ activity in cancer and normal samples of CRC patients as well as healthy patients. We showed that compared with healthy individuals, CRC patients have a higher incidence of *pks*+ E. coli mutational and indel signatures, and this is confirmed by metagenomics analysis on the same samples identifying the presence of *pks*+ genes. In addition, we demonstrated that both *pks*+ E. coli signature and short reads was found in both tumour and matching adjacent and distant normals in CRC patients. Results indicate that *pks*+ E. coli may be a significant driving force in the human gut since both the mutational signature and the homopolymer indel signature of pks+ are compatible with many driver mutations. These findings may perhaps represent additional factor potentially contributing to colon carcinogenesis, although further investigations in more controlled clinical settings are necessary to prove this mechanism. Since *pks*+ E.coli -induced mutagenesis occurs in the healthy colon of individuals without cancer, those individuals may be at an increased risk of developing CRC and hence *pks*+ E.coli may represent a potential biomarker of cancer risk.

## Methods
### Sample collection
Primary tumour tissue and matched normal samples were prospectively collected from patients undergoing curatively-intentioned surgery at University College London Hospital (UCLH). All patients gave informed consent for collection of their materials to the UCLH Cancer Biobank (REC approval 15/YH/0311). Four regions of each primary cancer were sampled by punch biopsy or scalpel dissection, at 12, 3, 6 and 9 o'clock positions around the tumour periphery. Tissue pieces were manually dissociated under the microscope using two 16 G needles, where individual glands were pulled away from the tissue mass.

### Sequencing and mutation calling
DNA fractions were extracted using the Zymo QuickDNA Microprep plus kit according to the manufacturer's instructions. Only samples with a total DNA yield higher than 10 ng were taken forward for WGS library preparation. Libraries were prepared using the NEBnext Ultra II FS kit according to manufacturer's instructions. Samples with sufficient library DNA yield and characteristic fragment size distribution (~200-500 bp) were further subjected to deep (~35x coverage) WGS.

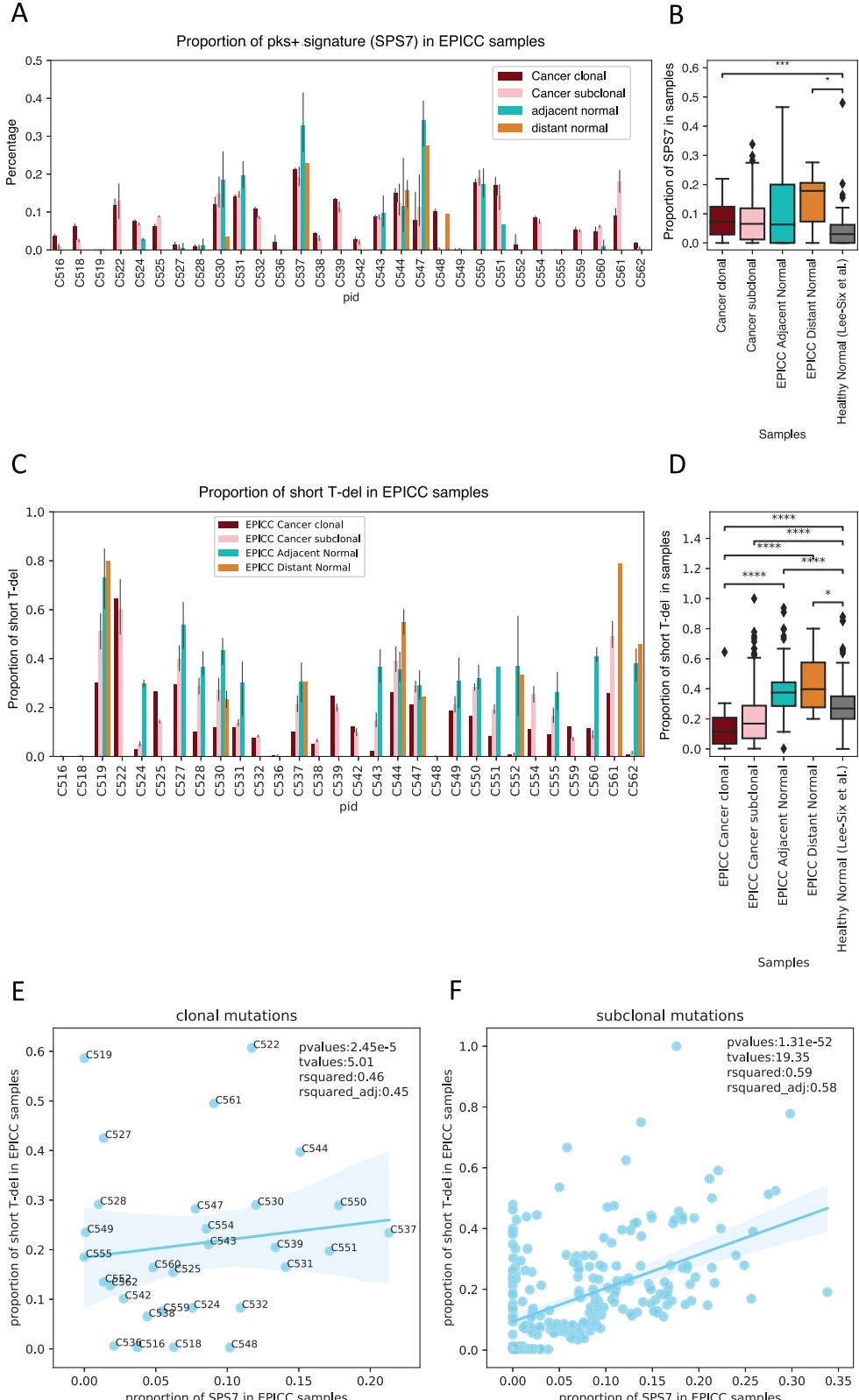

Sequence libraries were multiplexed and sequenced on an Illumina Novaseq.

The trimmed and filtered reads from each sequencing run and library where separately aligned to the GRCh38 reference assembly of the human genome[23] using the BWA-MEM algorithm v0.7.17[24] Following the GATK pipeline. Somatic mutations were first called with Mutect2. Somatic variants were annotated and candidate driver genes of colorectal cancers reported by3 and IntOGen34 as well as pan-cancer driver genes reported32 and81 filtered with the Variant Effect Predictor v93.282. To exclude contamination of a few cancer cells within the adjacent normal crypts, we removed any putative subclonal mutation in the sample and also excluded any somatic mutation that we also found in any of the corresponding cancer samples.

**Fig. 2 | *Pks*⁺ signature incidence. A, B** proportion of pks⁺ single base signature (SPS7, or COSMIC SBS88) in our dataset of 30 cancer patients and the comparison with normal crypts from normal people[8]. For each patient, there were 3-11 cancer crypts. And the number of patients that we had distant and adjacent normal crypts were 10 and 17, respectively. We also split the cancer clonal (dark red bars) and subclonal (pink bars) mutations when checking the signatures. **A** Data are presented as bars of mean ± SEM with single data points. **B** Box plots consist of the box denoting the interquartile range (IQR), bound by the 25th and 75th percentiles, the median line shown within the box, and the whiskers representing the rest of the data distribution with outliers denoted by points greater than ± 1.5 x IQR. Two-sided Mann–Whitney analysis was applied to compare groups. p($_{EPICC Cancer Clonal vs Healthy}$

$_{Normal}$) = 2.034e-02 ($n = 30$, $N = 40$), p($_{EPICC Distant Normal vs Healthy Normal}$) = 1.010e-04 ($n = 7$, $N = 40$). **C, D** proportion of short T-del signature at T-homopolymers in EPICC cancer and normal samples. D. Mann-Whitney-Wilcoxon test two-sided, p($_{EPICC Cancer Clonal vs EPICC Adjacent Normal}$) = 8.885e-11 ($n = 30$, $N = 71$), p($_{EPICC Cancer Clonal vs EPICC Distant Normal}$) = 7.776e-05 ($n = 30$, $N = 10$), p($_{EPICC Cancer Clonal vs Healthy Normal}$) = 1.805e-08 ($n = 30$, $N = 40$), p($_{EPICC Cancer Subclonal vs Healthy Normal}$) = 2.489e-16 ($n = 353, N = 30$), p($_{EPICC Adjacent Normal vs Healthy Normal}$) = 1.990e-08 ($n = 71$, $N = 40$), p($_{EPICC Distant Normal vs Healthy Normal}$) = 1.413e-02 ($n = 10$, $N = 40$). Correlation between SPS7 and proportion of short T-dels per sample in clonal (**E**) with Prob (F-statistic)= 2.45e-05 and subclonal (**F**) mutations with Prob (F-statistic)= 1.31e-52.

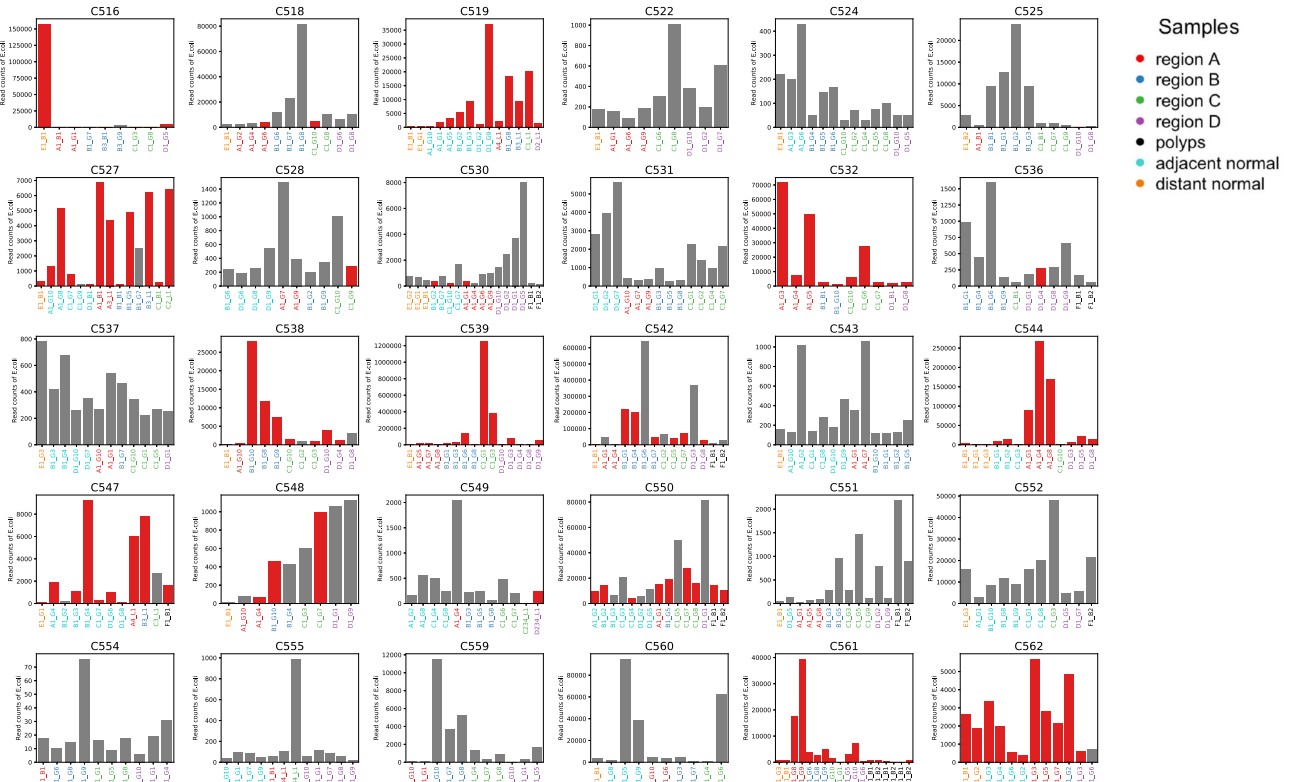

**Fig. 3 | *Pks*⁺ *E.coli* metagenomics.** The presence of *pks*+ genomic reads in the sequencing data of all the EPICC cohorts. Each panel present the samples from one patient, the x labels indicate the samples from different group distinguished by the colour (orange: distant normal crypts; cyan: adjacent normal crypts; others are the cancer crypts). The y-axis is the reads counts from *E.coli*. The red/grey color of the bars denotes the presence/absence of clb genes as the clb genes constitute pks genomic island and encoding colibactin.

## Phylogenetic reconstruction

Bayesian phylogenetic analyses of WGS data from EPICC cohort were performed using BEAST2[1]. The model we used was similar to the CRC phylogenetic analyses reported before[2]. We generated an input XML file for BEAST2 with BEAUti. The model and parameters we used are as below: as all clones were sampled at the same time, we set all tip dates as 0; as for the substitution model, we used the GTR model and set the Gamma Category Count to 4; we used the "Relaxed Clock Exponential" as clock model; As for prior for the relaxed clock rate mean, we used 4.6e-10 substitutions per site per generation[3]. As for Tree prior, we selected "Coalescent Exponential Population". Posteriors for the parameters of interest were obtained by running an MCMC chain during 100 million generations and sampled every 2000. We then constructed the maximum clade credibilty (MCC) tree using the TreeAnnotator[4]. In our study, we discarded the first 10% of the samples as burn-in and a maximum clade credibility topology was constructed using the median heights.

## Sparse signature identification

De novo mutational signatures extraction was performed with SparseSignatures[20]. This tool adopts LASSO regularisation to improve the fit, controlled by a regularisation parameter lambda (λ). It implements a scheme based on bi-cross-validation to estimate the optimal values for both the regularisation parameter λ and the number of signatures. We performed the inference considering a maximum of 10 signatures and scanning values of λ of 0.000, 0.025, 0.050 and 0.100. The best parameters were selected based on the median bi-cross-validation error estimated over 1000 iterations. This led to the de novo inference of 7 mutational signatures, which were then confirmed by a second analysis with SigProfiler using default parameters and a total of 1000 iterations.

We performed the inference on a joint cohort comprising tumour samples from our dataset of cancer patients and normal crypts from normal people[8]. We extracted trinucleotide counts for these samples in order to perform de novo mutational signatures extraction. In

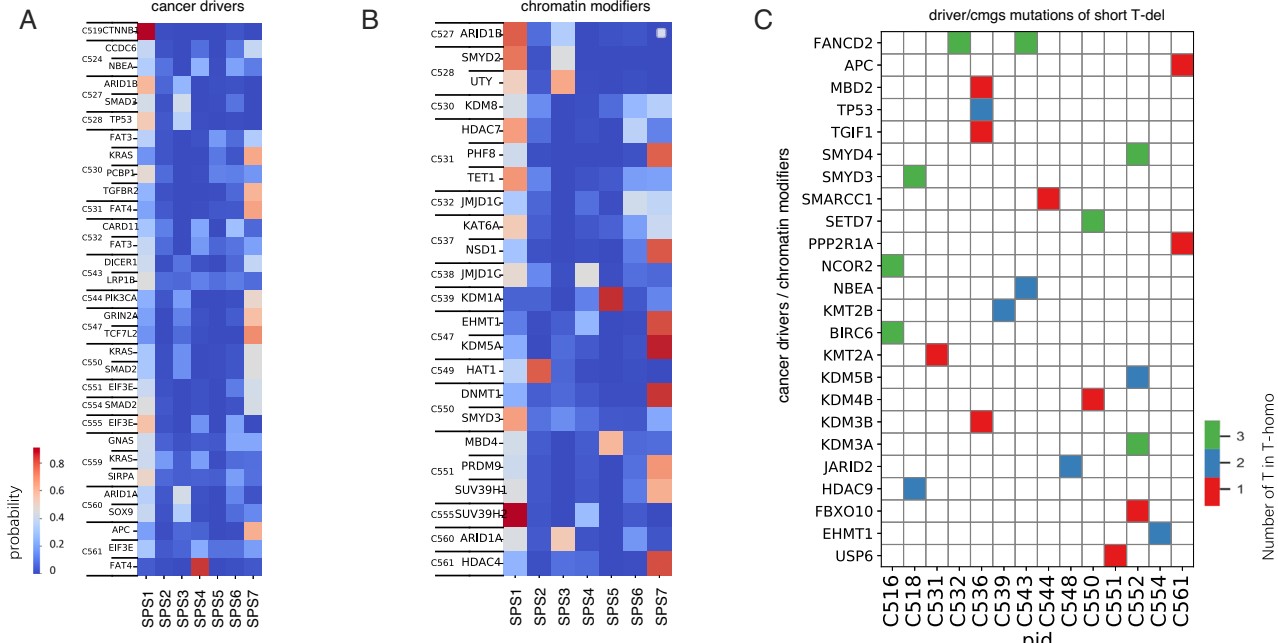

**Fig. 4 | Contribution of different mutational signatures to cancer driver mutations.** We estimated the probability that different signatures caused non-synonymous mutations in (**A**) cancer driver genes detected in tumour samples, as well as mutations in (**B**) chromatin modifier genes (cmgs). (**C**) Driver gene and chromatin modifier gene alterations caused by short T-dels, likely by pks + .

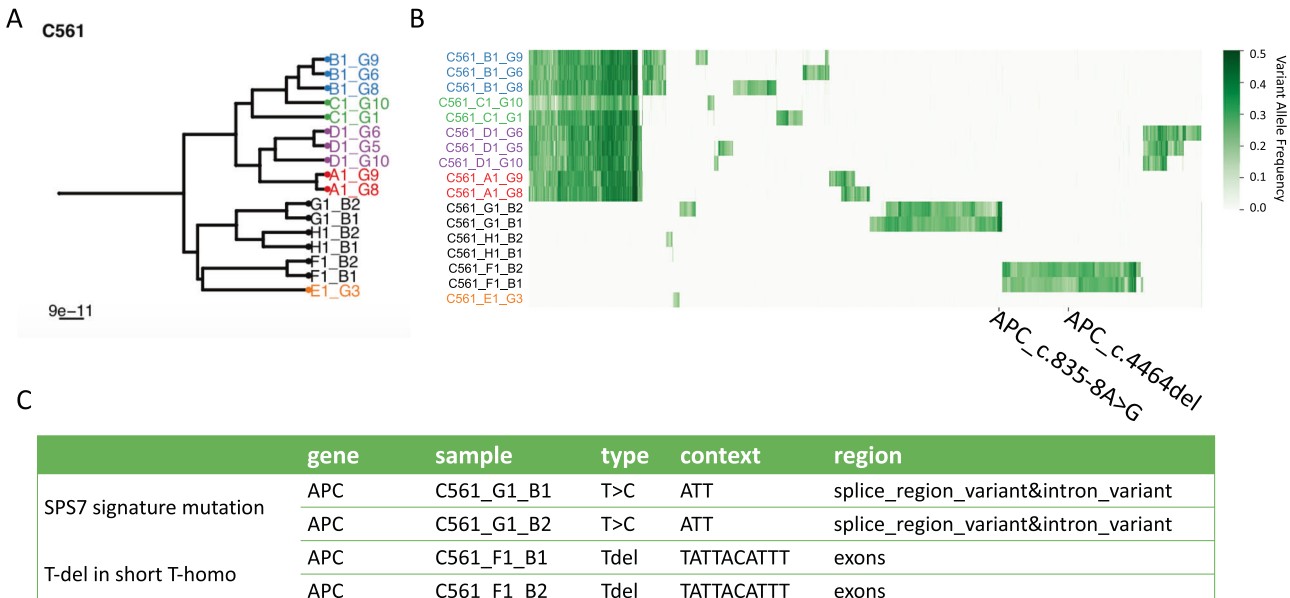

| | gene | sample | type | context | region |
|---|---|---|---|---|---|
| SPS7 signature mutation | APC | C561_G1_B1 | T>C | ATT | splice_region_variant&intron_variant |
| | APC | C561_G1_B2 | T>C | ATT | splice_region_variant&intron_variant |
| T-del in short T-homo | APC | C561_F1_B1 | Tdel | TATTACATTT | exons |
| | APC | C561_F1_B2 | Tdel | TATTACATTT | exons |

**Fig. 5 | Showcase of patient C561. A** The phylogeny of all samples for patient C561, the black labels are the independent precancerous polys lesions. **B** The heatmap of SNVs for C561 (**C**) APC driver mutations found in polyps samples matching the pks⁺ single base signature (polyp G) and short T-del signature motifs (polyp F).

particular, all point mutations are grouped into six categories: C > A, C > G, C > T, T > A, T > C, and T > G, where the original pyrimidine base is listed first. Next, these categories are further divided into 96 sub-categories based on the 16 possible combinations of 5' and 3' flanking bases. Each sample is characterized by the number of mutations in each of these 96 subcategories. This information is represented in a count matrix, where the rows correspond to samples and the columns represent the 96 subcategories. The goal of de novo mutational signatures extraction is to factorise such count matrix into the product of two matrices with low ranks: the exposure matrix, consisting of one row per tumour and K columns, and the signature matrix, with K rows and 96 columns. Here, K denotes the number of signatures.

We employed SparseSignatures[20] to conduct de novo extraction of mutational signatures. This approach involves two main steps. First, an initial inference step utilises Non-Negative Matrix Factorisation to minimise the squared residual error between observed counts and their predicted counterparts, while ensuring that all elements remain non-negative. Subsequently, the inferred signatures undergo refinement through LASSO regularisation. This regularisation technique effectively mitigates overfitting by employing an L1 penalty controlled

by a regularisation parameter lambda (λ). It is important to highlight that the objective function we minimise in this context is non-convex, in contrast to the standard LASSO. However, it exhibits bi-convexity, meaning it is convex when one matrix is fixed while optimising the other, and vice versa. As a result, we employ an alternating learning algorithm for the inference process, where we iteratively update one matrix while keeping the other fixed, and then switch roles. This iterative procedure is repeated multiple times. Previous studies[20] have demonstrated that convergence is typically achieved within 10 to 20 steps. Therefore, conservatively, we performed a total of 20 iterations to ensure a thorough exploration of the solution space.

SparseSignatures[20] incorporates a bi-cross-validation scheme to estimate the optimal values for both the regularisation parameter λ and the number of signatures K. This approach involves multiple independent runs of bi-cross-validation, wherein 1% of the cells of the input counts matrix is randomly selected and set to zero. Different values of λ and K are tested, and the de novo inference is executed for each configuration. The mean bi-cross-validation error, which quantifies the discrepancy between the true and predicted cells among the removed ones, is computed for each configuration. The values of λ and K that minimise the mean bi-cross-validation error are selected as the optimal choices[20].

We performed the inference considering a maximum of 10 signatures and scanning values of λ of 0.000, 0.025, 0.050 and 0.100. The best parameters were selected based on the median bi-cross-validation error estimated over 1,000 iterations. This led to the de novo inference of 7 mutational signatures. To computationally validate the identified signatures, a second analysis was conducted using SigProfiler. This method is also based on Non-Negative Matrix Factorisation, similarly to the previous approach, and incorporates two metrics for determining the optimal number of signatures: stability of inference across multiple runs and goodness of fit measured by mean squared error. SigProfiler was executed with default parameters, and a rigorous iteration of 1000 runs was performed to ensure robustness and accuracy in the analysis. The signatures obtained through SigProfiler exhibited a high degree of consistency with those inferred using SparseSignatures[20]. The agreement between the two methods further strengthens the confidence in the identified signatures and their relevance in capturing the underlying mutational processes."

### ID signature

To obtain counts data for Small Insertions and Deletions (ID) signatures, we considered the 83 mutation types defined in COSMIC (https://cancer.sanger.ac.uk/signatures/id/). These counts were derived by taking into account the size of the indels, the nucleotides affected, and whether they were present in repetitive or microhomology regions. We utilised the COSMIC catalogue, which consists of 18 ID signatures, to assign signatures to the samples. This assignment process involved minimising the mean squared error between the observed counts and the predicted counts, with the implementation of the LASSO L1 penalty to mitigate overfitting and improve the accuracy of the signature assignment.

### *E.coli* reads extraction

The bioinformatics pipeline PathSeq[25] was used to identify *pks⁺ E. coli* sequences in data from colorectal cancer and normal samples. First, we subtracted the sequencing reads from human genome and then align the remaining(non-host) reads to the *pks⁺ E. coli* genome. From the output high-quality non-host reads aligned to *pks⁺ E. coli* reference, we extracted ClbA-ClbS genes as these genes encode *polyketide synthases*(PKS).

### Reporting summary

Further information on research design is available in the Nature Portfolio Reporting Summary linked to this article.

## Data availability

Analysed data (both for the original EPICC dataset and the new normal crypt data) are openly available on Mendeley: [https://data.mendeley.com/datasets/dvv6kf856g/2]. Raw sequencing data (both for the original EPICC cohort and the new normal crypt data) have been deposited at the European Genome-phenome Archive (EGA), which is hosted by the EBI and the CRG, under accession number EGAS00001005230. Further information about EGA can be found on [https://web2.ega-archive.org/about/introduction]. Due to the personal nature of sequencing data, access to these data is restricted and subject to application. Access will be granted for the duration of the proposed project.

## Code availability

Complete scripts to replicate all bioinformatic analysis and perform simulations and inference are available at: https://github.com/sottorivalab/EPICC2021_data_analysis.

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

## Acknowledgements
We acknowledge support from the Medical Research Council (MR/P000789/1) to A.S. A.S. and T.G. are supported by the Wellcome Trust (202778/B/16/Z and 202778/Z/16/Z respectively) and Cancer Research UK (A22909 to A.S., A19771 and DRCNPG-May21_100001 to T.G.). This work was also supported by the CRUK Accelerator Award (A26815) and the Wellcome Trust award to the Centre for Evolution and Cancer (105104/Z/14/Z). We acknowledge funding from the National Institute of Health (NCI U54 CA217376) to A.S. Figure 1A-C was created with BioRender.com.

## Author contributions
B.C. analysed and interpreted the data. D.R. performed mutational signature analysis and interpretation. T.H. analysed the data. I.S and J.F. collected the samples and generated the data. C.J. contributed to data analysis. L.M. contributed to result interpretation. T.A.G. and A.S. conceived, obtained funding, supervised the study and wrote the manuscript.

## Competing interests
The authors declare no competing interests.
