## [Peer Review File · Nature Communications]

REVIEWER COMMENTS

Reviewer #1 (Remarks to the Author):

In their resubmission, Chen et al. present a number of new discussion and analyses that adequately address the concerns I had with their original manuscript. Namely, a more thorough framing for the novelty and impact of this paper, the inclusion of graphs correlating the presence of SPS7 with short T-dels in the same sample, the inclusion of analyses demonstrating the correlation of SPS7 with pks metagenomic reads in MSS/MSI patients, and a more in-depth analysis of ID signatures as they relate to short T-dels.

While I feel that the authors do a good job presenting their study clearly and make several impactful points, there are some nuances to how the authors reference various genomic signatures that I found somewhat confusing or inconsistent. My response to each of the authors comments is listed below:

Comment 1.1: The authors have done a good job addressing this critique and more clearly explaining the novelty of this study.

Comment 1.2: The assertion that MSI signatures or past exposure to pks metagenomic reads may mask or underestimate the contribution of pks-associated mutational signatures is well made. The addition of Figures 2E,F demonstrating the correlation between short T-dels and SPS7 strengthens the case for colibactin-driven tumorigenesis, and the inclusion of Figure S6 convincingly shows the correlation of metagenomic pks reads with short T-dels in MSS patients. However, there are some limitations here that I will discuss more in comment 1.4.

Comment 1.3: The authors expansion to include all 17 pks-colonized patients satisfies this critique. Moreover, the included consideration for short T deletions in driver or cmg genes is a nice addition.

Comment 1.4: While I agree with the authors assessment that the extraction of indel signatures is bioinformatically complex, this signature has been extracted by previous groups from CRC tumor

biopsies and shown to correlate with the pks SBS signature. Of course, this is not to say that the limitations the authors propose is not valid, or that the short-T signature observed has no biological relevance to colibactin. However, it is confusingly introduced in the paper as the canonical pks ID signature (ID-pks): “We then investigated whether ID-pks (short T deletions at T homopolymers) signatures occur in both normal and cancer crypts of our dataset. The ID-pks signature was more evident than SBS7”. Later (red-text) this is referred to simply as a “short T-del signature”. The latter wording is more accurate, as what is being detected is not canonically the ID-pks signature that has been previously described. Regardless of the semantics, I think the authors point regarding the prevalence of short T-dels is clear and supported by evidence, but the nomenclature needs to be consistent (not ID-pks)

Comment 1.5: The addition of cosine similarities between the inferred signatures and those in the COSMIC database resolves this comment.

Reviewer #2 (Remarks to the Author):

As I have already reviewed a previous version of this article, I will comment mostly on the authors' changes and replies to my previous requests.

I would like to thank the authors for the changes from the previous submission, which improved the manuscript presentation and clarity. At the same time, I believe some additional work is required before I can advise publication.

Major:

1. I think it would be important to specify in the abstract and at least at the beginning of the results section, a summary of the samples analysed so that it is clear: how many samples are from normal tissue of cancer patients and how many are from tumours, and what type of samples they are (bulk/single clone, WGS deep/shallow). Perhaps a Figure panel would help to summarise the dataset. Also, the authors should mention at least in the text, how many samples are new and how many are from previously published studies.
2. Despite some improvements on the presentation of the samples, it is still confusing to me to look at some of the figures because of the unclear number of samples/patients and the sample name notation. For example, reaching Figure 1D, we see there are 40 samples there, but it is unclear whether these are all the samples, and which samples are from the same patient. I could not find the explanation, but I assume that the CXXX number refers to the patient, but it is hard to visually group them together because of small text and long sample names. The sample naming is also not sufficiently explained and does not allow the reader to distinguish between cancer and normal adjacent to cancer samples. The

legend states “the crypts (both adjacent normal and cancer) from different regions of cancer tissue were marked as ‘A_’, ‘B_’, ‘C_’, ‘D_’, while ‘E_’ are the crypts from distant normal tissues.” However, sample names in Figure 1D read for example EPICC_C519_D1_G2_D1. What do the other letters refer to? Is this a cancer sample or an adjacent normal? I suggest to 1) clarify the number of samples and inclusion criteria, 2) improve the figures so that samples from the same patient are easier to identify, and 3) samples that are from cancer can be distinguished from samples that are from adjacent normal, and 4) change or fully explain sample naming.

3. While the authors mention that now Figure 3 shows data from all patients, the updated manuscript still shows only 25/30 patients. Please update adding patients or specify inclusion criteria. For Figure 4, more patients have been added, but the inclusion criteria are still missing, please add it and revise all figures so that the inclusion criteria are specified for both samples and patients.

4. My suggestions for strengthening the mutational signature analysis were mostly ignored. Despite additional explanation of the methods, one point I made was not fully addressed. In a previous analysis of the same cancer samples, Heide et al 2022, the colibactin signature SBS88 was not identified, while in the normal samples of Lee-Six 2019, SBS88 was indeed extracted. So, it is not surprising that combining the data from the two studies then SBS88 is found. However, I think more needs to be done to demonstrate that SBS88 is indeed present in the cancer samples, and not just overfitted, or “bleeding out” from Lee-Six data into Heide data. There are I think a few possible ways to prove that SBS88 is in Heide data: 1) perform another signature extraction on the Heide data only, and show that SBS88 can be extracted; 2) fit Heide data with the original signatures and demonstrate that some samples cannot be fully explained by the original signatures and that adding SBS88 significantly improves the fit; 3) show examples of mutational catalogues of samples in Heide data that have a sufficiently high contribution of SBS88 that the signature can be undeniably observed in these samples. Moreover, if the number of samples is sufficient, I would like to see separate signature extractions for 1) cancer samples, 2) non-cancer adjacent, 3) non-cancer distant. At least show that it was attempted and what the limitations are. I am particularly concerned about overfitting, because of what I mentioned in my previous review, where I noticed that the mismatch repair deficiency signature SPS6 (SBS44) was assigned to most patients in small proportions, while it is expected to be present only in the few MSI tumours.

Minor:

5. Signature naming consistency. I noticed that the authors seem to use the abbreviation SBS and SPS interchangeably when referring to the signatures identified in this study. This creates some confusion, because it is not clear whether they refer to their own numbering system or the established COSMIC numbering. For example, in the COSMIC numbering, the colibactin signature is SBS88, while SBS7 is the UV light signature, however the authors refer to the colibactin signature as both SBS7 and SPS7. I suggest revising for clarity, using SBS with the COSMIC numbering and SPS to their own numbering. It would help to explain at the beginning of the article that SPS are the signatures obtained with SparseSignatures, and that each of the SPS1-SPS7 corresponds to a certain COSMIC SBS signature. Then, in the remainder article the authors can use either the SPS or SBS numbers (for example either call the colibactin signature SPS7 or SBS88), and possibly use only one of the two notations to avoid the confusion of using two naming conventions at the same time.

6. Figure 1D, what does red and blue indicate? Please add a legend.

We thank the reviewers once again for assessing our revised version of the manuscript and providing constructive feedback. Here we address their final concerns in a point-by-point format.

Reviewer #1:

In their resubmission, Chen et al. present a number of new discussion and analyses that adequately address the concerns I had with their original manuscript. Namely, a more thorough framing for the novelty and impact of this paper, the inclusion of graphs correlating the presence of SPS7 with short T-dels in the same sample, the inclusion of analyses demonstrating the correlation of SPS7 with pks metagenomic reads in MSS/MSI patients, and a more in-depth analysis of ID signatures as they relate to short T-dels.

While I feel that the authors do a good job presenting their study clearly and make several impactful points, there are some nuances to how the authors reference various genomic signatures that I found somewhat confusing or inconsistent. My response to each of the authors comments is listed below:

We thank the reviewer for appreciating our efforts and apologize for not fully addressing their comments in the first instance.

Comment 1.1: The authors have done a good job addressing this critique and more clearly explaining the novelty of this study.

Thank you.

Comment 1.2: The assertion that MSI signatures or past exposure to pks metagenomic reads may mask or underestimate the contribution of pks-associated mutational signatures is well made. The addition of Figures 2E,F demonstrating the correlation between short T-dels and SPS7 strengthens the case for colibactin-driven tumorigenesis, and the inclusion of Figure S6 convincingly shows the correlation of metagenomic pks reads with short T-dels in MSS patients. However, there are some limitations here that I will discuss more in comment 1.4.

This is a good point that indeed we address in comment 1.4.

Comment 1.3: The authors expansion to include all 17 pks-colonized patients satisfies this critique. Moreover, the included consideration for short T deletions in driver or cmg genes is a nice addition.

Thanks.

Comment 1.4: While I agree with the authors assessment that the extraction of indel signatures is bioinformatically complex, this signature has been extracted by previous groups from CRC tumor biopsies and shown to correlate with the pks SBS signature. Of course, this is not to say that the limitations the authors propose is not valid, or that the short-T signature observed has no biological relevance to colibactin. However, it is confusingly introduced in the paper as the canonical pks ID signature (ID-pks): "We then investigated whether ID-pks (short T deletions at T

homopolymers) signatures occur in both normal and cancer crypts of our dataset. The ID-pks signature was more evident than SBS7". Later (red-text) this is referred to simply as a "short T-del signature". The latter wording is more accurate, as what is being detected is not canonically the ID-pks signature that has been previously described. Regardless of the semantics, I think the authors point regarding the prevalence of short T-dels is clear and supported by evidence, but the nomenclature needs to be consistent (not ID-pks).

We apologise with the lack of clarity; we have now amended the revised manuscript to reflect the suggestions of this reviewer (text in red).

Comment 1.5: The addition of cosine similarities between the inferred signatures and those in the COSMIC database resolves this comment.

Thanks.

Reviewer #2 (Remarks to the Author):

As I have already reviewed a previous version of this article, I will comment mostly on the authors' changes and replies to my previous requests.

I would like to thank the authors for the changes from the previous submission, which improved the manuscript presentation and clarity. At the same time, I believe some additional work is required before I can advise publication.

We are thankful to the reviewer for evaluating our revised version of the manuscript and we are sorry we did not fully address their comments in the first iteration, but we are confident we did it now (see below).

Major:

1. I think it would be important to specify in the abstract and at least at the beginning of the results section, a summary of the samples analysed so that it is clear: how many samples are from normal tissue of cancer patients and how many are from tumours, and what type of samples they are (bulk/single clone, WGS deep/shallow). Perhaps a Figure panel would help to summarise the dataset. Also, the authors should mention at least in the text, how many samples are new and how many are from previously published studies.

This is a fair point, we have now applied those suggestions to the revised version of the manuscript and added a new panel in Figure 1E.

2. Despite some improvements on the presentation of the samples, it is still confusing to me to look at some of the figures because of the unclear number of samples/patients and the sample name notation. For example, reaching Figure 1D, we see there are 40 samples there, but it is unclear whether these are all the samples, and which samples are from the same patient.

We apologise again for the lack of clarity. In this figure panel we report only those normal crypt samples where we find a mutation in a cancer driver gene. We now

specify this detail in the figure legend and text of the revised manuscript, and we have also added a new Supplementary Figure 1 with all the samples (showing that the rest of the samples have no cancer driver mutation).

I could not find the explanation, but I assume that the CXXX number refers to the patient, but it is hard to visually group them together because of small text and long sample names. The sample naming is also not sufficiently explained and does not allow the reader to distinguish between cancer and normal adjacent to cancer samples. The legend states “the crypts (both adjacent normal and cancer) from different regions of cancer tissue were marked as ‘A_’, ‘B_’, ‘C_’, ‘D_’, while ‘E_’ are the crypts from distant normal tissues.” However, sample names in Figure 1D read for example EPICC_C519_D1_G2_D1. What do the other letters refer to? Is this a cancer sample or an adjacent normal? I suggest to 1) clarify the number of samples and inclusion criteria, 2) improve the figures so that samples from the same patient are easier to identify, and 3) samples that are from cancer can be distinguished from samples that are from adjacent normal, and 4) change or fully explain sample naming.

We are sorry for the confusion; indeed we do agree that the sample naming is really not easy to follow. For further analyses and for reusability and trackability of the data (including the raw data we will share with the community), we opted to avoid changing sample naming as this may risk losing track of samples within the study. We instead added a naming legend to Figure 1A and explain better the naming convention in the revised manuscript (text in red). We hope now the naming is much clearer.

3. While the authors mention that now Figure 3 shows data from all patients, the updated manuscript still shows only 25/30 patients. Please update adding patients or specify inclusion criteria. For Figure 4, more patients have been added, but the inclusion criteria are still missing, please add it and revise all figures so that the inclusion criteria are specified for both samples and patients.

Apologies, we have now included all samples.

4. My suggestions for strengthening the mutational signature analysis were mostly ignored. Despite additional explanation of the methods, one point I made was not fully addressed. In a previous analysis of the same cancer samples, Heide et al 2022, the colibactin signature SBS88 was not identified, while in the normal samples of Lee-Six 2019, SBS88 was indeed extracted. So, it is not surprising that combining the data from the two studies then SBS88 is found. However, I think more needs to be done to demonstrate that SBS88 is indeed present in the cancer samples, and not just overfitted, or “bleeding out” from Lee-Six data into Heide data. There are I think a few possible ways to prove that SBS88 is in Heide data: 1) perform another signature extraction on the Heide data only, and show that SBS88 can be extracted; 2) fit Heide data with the original signatures and demonstrate that some samples cannot be fully explained by the original signatures and that adding SBS88 significantly improves the fit; 3) show examples of mutational catalogues of samples in Heide data that have a sufficiently high contribution of SBS88 that the signature can be undeniably observed in these samples. Moreover, if the number of samples is sufficient, I would like to see separate signature extractions for 1) cancer samples, 2)

non-cancer adjacent, 3) non-cancer distant. At least show that it was attempted and what the limitations are. I am particularly concerned about overfitting, because of what I mentioned in my previous review, where I noticed that the mismatch repair deficiency signature SPS6 (SBS44) was assigned to most patients in small proportions, while it is expected to be present only in the few MSI tumours.

We apologise for the incomplete response to this reviewer's point. We followed the advice above and first performed signature deconvolution separately for the three groups of samples: distant normals, adjacent normals and cancer (as previously done in Heide et al. 2022). We indeed were able to identify the *pks+* *E.coli* signature in both normal cohorts (Figure R1A,B and new Supplementary Figure XA, B), but not in the cancer samples alone, where the dominant signatures were those reported in Heide et al. 2022 (Figure R1C and new Supplementary Figure XC). We then analysed the cancer cohort more in depth, and compared the cosine similarities of signature extraction with and without the *pks+* *E.coli* signature, hence assigning only SPS1-6 versus assigning SPS1-7. The analysis showed that the exclusion of SPS7 resulted in a significant decrease in the goodness of fit (Wilcoxon test p-value=0.033, as shown in Figure R1D, new Supplementary Figure 5D). These results provide evidence for the significant impact of SPS7 on the mutations observed in cancer patients as well, highlighting the difficulties of inferring robust mutational signatures.

Figure R1. Mutational signature decomposition divided by group of samples.

Minor:

5. Signature naming consistency. I noticed that the authors seem to use the abbreviation SBS and SPS interchangeably when referring to the signatures identified in this study. This creates some confusion, because it is not clear whether they refer to their own numbering system or the established COSMIC numbering. For example, in the COSMIC numbering, the colibactin signature is SBS88, while SBS7 is the UV light signature, however the authors refer to the colibactin signature as both SBS7 and SPS7. I suggest revising for clarity, using SBS with the COSMIC numbering and SPS to their own numbering. It would help to explain at the beginning

of the article that SPS are the signatures obtained with SparseSignatures, and that each of the SPS1-SPS7 corresponds to a certain COSMIC SBS signature. Then, in the reminder article the authors can use either the SPS or SBS numbers (for example either call the colibactin signature SPS7 or SBS88), and possibly use only one of the two notations to avoid the confusion of using two naming conventions at the same time.

Sorry for that, we have clarified and corrected our references to the signatures.

6. Figure 1D, what does red and blue indicate? Please add a legend.

Apologies, we have now added the legend.

REVIEWERS' COMMENTS

Reviewer #1 (Remarks to the Author):

The authors have made appropriate revisions to address comment 1.4 and I have no further comments.

Reviewer #2 (Remarks to the Author):

I believe the authors have addressed all the concerns I raised during the previous two rounds of review and I am happy to advise publication.